# INFORMATION SPREADING IN DIFFUSION MODELS FROM EFFECTIVE FIELD THEORY

## ABSTRACT

We study score-matching diffusion models with a convolutional architecture. We argue that the inductive bias of locality means that the machinery of *effective field theory* from physics can be usefully applied to describe the denoising dynamics. As a simple application we study a toy example and show that the mutual information between two points grows in a manner predicted by a simple effective field theory. We also show the same result holds when trained on MNIST.

## 1 MOTIVATION

In physics, it is often possible to quantitatively describe complicated phenomena using simplified models. Basic principles such as locality and symmetry allow *effective field theories* to be built, where only a small number of parameters control the long-distance and late-time behavior. In this work we seek to use this approach to describe the score function in a diffusion model, and make sharp quantitative statements about the dynamics during its reverse process. We hope that eventually this approach will allow for deeper insights beyond our investigation here.

### 1.1 PREVIOUS WORK

Diffusion models Sohl-Dickstein et al. (2015) Nichol & Dhariwal (2021) are a powerful and commonly used method in generative AI. In this work we study these models in their score-matching incarnation Song et al. (2021). Previous work on understanding general principles underlying these models from a physics-inspired perspective includes Raya & Ambrogioni (2023),Cotler & Rezchikov (2023), Stancevic et al. (2025). The regimes we describe in the reverse process are similar to those detailed in Biroli et al. (2024), and our work aims to better understand the 'Brownian motion' regime therein. Information theory was applied to study mutual information in the reverse process in Guo et al. (2004); Venkat & Weissman (2012); Kong et al. (2023), where we use related tools to study specifically spatial correlations at the pixel level.

Effective field theory (EFT) is a classic approach in physics Weinberg (1979); Polchinski (1992) where constraints on the description of a system - in physics, typically locality, symmetry or causality - allow for the construction of simplified models that describe its dynamics. We apply this approach to model the reverse process of a diffusion model, to understand how mutual information is created and spreads in the sample as it is gradually denoised.

## 2 THEORY

Recall that diffusion models first gradually noise training data samples to noise in the forward process, and from this then learn the correct denoising pattern in the reverse process. The form of the reverse process we use[1], for field $\phi$, is the deterministic ODE form without noise:

$$\frac{\partial \phi(t)}{\partial t} = \frac{\partial_t \bar{\alpha}_t}{\bar{\alpha}_t} \left[ \phi(t) + \pi\left(\phi(t), t\right) \right] \tag{1}$$

This is the ODE form of the reverse process as found in Song et al. (2021). Here, $\pi(\phi(t), t)$ is the score function learned by the neural network. The noise scheduling function is $\bar{\alpha}_t$ where $\bar{\alpha}_T = 0$ at the pure noise endpoint (beginning of reverse process), and $\bar{\alpha}_0 = 1$, at the data endpoint (end of reverse process).

In this work we will consider data that is defined on $\mathbb{R}^2$ (e.g. images) so $\phi = \phi(x, t)$. We will also consider denoising neural networks with a convolutional architecture. It has been argued by Kamb & Ganguli (2025) that the inductive bias associated with this locality can be used to understand

---

[1] Our conventions are those of Kamb & Ganguli (2025).

a form of creativity shown by the diffusion model. In particular, the empirical form of the score function is called the Equivariant Local Score (ELS) Machine, and is formulated in terms of local patches $\phi_{\Omega_x}$ centred at $x$, and training data patches $\varphi$:

$$M_t[\phi](x) = -\frac{1}{1-\bar{\alpha}_t}\frac{\sum_{\varphi \in P_\Omega(\mathcal{D})}\left(\phi(x)-\sqrt{\bar{\alpha}_t}\varphi(0)\right)\mathcal{N}(\phi_{\Omega_x}|\sqrt{\bar{\alpha}_t}\varphi,(1-\bar{\alpha}_t)I)}{\sum_{\varphi' \in P_\Omega(\mathcal{D})}\mathcal{N}(\phi_{\Omega_x}|\sqrt{\bar{\alpha}_t}\varphi'|(1-\bar{\alpha}_t)I)} \tag{2}$$

In this work we further explore the consequences of this locality. In particular, if the network has a finite receptive field as described in Luo et al. (2016), then we can attempt to apply principles of *effective field theory* Weinberg (1979) from physics. Any local functional of the field $\phi(x,t)$ can *in principle* be expanded in derivatives, i.e. we can consider writing the score on the right hand side of (1) as an expansion of the following form:

$$\frac{\partial \phi(x,\bar{\alpha}_t)}{\partial \bar{\alpha}_t} = g_0(\bar{\alpha}_t) + g_2(\bar{\alpha}_t)\mathbf{v}_2 \cdot \nabla\phi(x,t) + g_3(\bar{\alpha}_t)\left[\nabla^T \mathbf{M}_3 \nabla\right]\phi(x,t)+$$

$$+ g_4\phi^2(x,t) + g_5\phi^2(x,t)\mathbf{v}_5 \cdot \nabla\phi(x,t) + g_6(\bar{\alpha}_t)[\nabla\phi(x,t)]^T\mathbf{M}_6\nabla\phi(x,t) + ... \tag{3}$$

where we have chosen to use $\bar{\alpha}_t$ as the time coordinate in (1), and where we have chosen to perform a double expansion in $\phi$ and in its derivatives. Here the $\mathbf{v}_i$ are constant vectors, and the $\mathbf{M}_i$ matrices, which are determined in terms of the underlying dataset. We have also performed a field redefinition to simplify the expansion slightly; see Appendix A.1 for details.

Given a precise microscopic description – which is quite rare in machine learning, but in this case has been argued to exist by the ELS – we can in principle provide explicit expressions for all the coefficients in the above expansion, which we do for the first few of the ELS in Appendix A.2

Why would such a formal expansion be useful? In general it is often possible to ignore all but a finite subset of the terms here and focus only on the terms that are lowest-order in derivatives. Thus only a few parameters control observables, allowing for precise predictions.

## 2.1 MUTUAL INFORMATION GROWTH

The observable that we are interested in is the mutual information between two points in the image; at the beginning of the reverse process this is zero as each point is uncorrelated to the rest. We will argue that its behavior can display universal behavior governed by an EFT. We first define the connected two-point correlation function of the samples $\phi$ at time $t$ during the reverse process:

$$G_t(x,y) := \mathbb{E}\left[\phi(x,t)\phi(y,t)\right] - \mathbb{E}\left[\phi_t\right]^2 \tag{4}$$

where the angled brackets denote an ensemble average (in this case, over multiple realisations of the initial noise).

From this, under the assumption of Gaussian evolution, the mutual information between the lattice points $\phi(x)$ and $\phi(y)$ may be written

$$I(x,y;t) = -\frac{1}{2}\log\left[1-\rho(x,y;t)^2\right] \tag{5}$$

where $\rho(x,y;t)$ is the correlation coefficient of the distributions of the two pixels at time $t$:

$$\rho(x,y;t) := \frac{G_t(x,y)}{\mathbb{E}\left[\phi_t^2\right] - \mathbb{E}\left[\phi_t\right]^2} \tag{6}$$

We shall calculate the mutual information $I(x,y;t)$ from $\rho(x,y;t)$ in our example cases, assuming approximate Gaussianity during the diffusive regime we are interested in.

## 2.2 SNAPPING REGIME AT END OF REVERSE PROCESS

Near the beginning of the reverse process, mutual information is created and structure emerges. Near the end, however, this no longer happens, no new structure is being formed, and patches instead 'snap' to their nearest training data examples. Here, where $\bar{\alpha}_t \to 1^-$, a different expansion holds. We shall expand the ELS in $\epsilon_t := 1 - \bar{\alpha}_t$, and separate out the patch nearest to $\phi$, calling it $\varphi^*$.

$$\frac{\partial \phi(x)}{\partial t} = -\frac{\partial_t \epsilon_t}{1-\epsilon_t}\frac{1}{\epsilon_t}\left((1-\epsilon_t)\phi(x) - \sqrt{1-\epsilon_t}\frac{\varphi^*(0) + \sum_{\varphi \in P_\Omega(\mathcal{D})}\varphi(0)e^{(\phi_{\Omega_x}-\sqrt{\bar{\alpha}_t}\varphi^*)^2 - (\phi_{\Omega_x}-\sqrt{\bar{\alpha}_t}\varphi)^2/2\epsilon_t I}}{1 + \sum_{\varphi' \in P_\Omega^y(\mathcal{D})}e^{(\phi_{\Omega_x}-\sqrt{\bar{\alpha}_t}\varphi^*)^2 - (\phi_{\Omega_y}-\sqrt{\bar{\alpha}_t}\varphi')^2/2\epsilon_t I}}\right) \tag{7}$$

Thus, simplifying and dropping subleading terms in $\epsilon$ - which here are both $\mathcal{O}(\epsilon_t)$ and $\mathcal{O}(e^{-1/\epsilon_t})$ - we get a leading order evolution equation governing the end of the reverse process:

$$\frac{\partial \phi(x)}{\partial t} \approx -\frac{\partial_t \epsilon_t}{\epsilon_t} \left( \phi(x) - \varphi^*(0) \right) \tag{8}$$

This equation describes a straightforwardly calculable trajectory to the nearest training patch $\varphi^*$.

The same idea is used in Theorem 4.1 of Kamb & Ganguli (2025) to establish locally consistent points. We have here however presented an explicit evolution equation that describes the end of the reverse process.

## 3 EXPERIMENT ON BLACK AND WHITE GRIDS

We shall now demonstrate an effective field theory regime quantitatively in the simplest possible case with a full analytical description.

Following Kamb & Ganguli (2025), we train a simple convolutional neural network for a diffusion model with only two samples: a grid where every pixel has the value $1$, and another grid where every pixel is $-1$. Due to the locality bias of the ConvNet, the diffusion model will generically produce patches with value $1$ and $-1$, of a characteristic size smaller than the size of the whole grid, as seen in Figure 1. This was argued to be a simple example of the "creativity" described by the diffusion model: it glues together patches of the original training data. We will study both the ELS (which provides an idealized description of the dynamics of the diffusion model), and the trained diffusion model itself: we will see that the results are extremely similar.

We can now attempt to apply the formalism of the effective field theory to this case. Let us look near the pure noise endpoint (where $\bar{\alpha}_t \approx 0$ as $t \to 0^+$). This means that $\bar{\alpha}_t$ itself is a perturbative parameter, and we need only look at leading order in it - in this case, $\mathcal{O}(1)$ (see Appendix A.2). Furthermore, locality means that terms in the evolution equation can only depend on a combination of the field $\phi$ or its gradients $\nabla^n \phi$ for any $n$.

In this case, the rotational symmetry of the dataset means that any linear gradient term $\propto \nabla \phi$ can be set to $0$, while the only second derivative term must be the rotationally invariant Laplacian $\nabla^2$. for details of this derivation see Appendix A.2. Meanwhile, the dataset has $0$ mean, and is invariant under sign reversal, meaning that no linear terms $\propto \phi$ can appear in the EFT that change the mean of $\phi$.

The first contributing term is then $\nabla^2 \phi$, with which we find a diffusive regime of the form

$$\frac{\partial \phi(x, \bar{\alpha}_t)}{\partial \bar{\alpha}_t} \approx c_3 \nabla^2 \phi(x, t) \tag{9}$$

The reasoning justifying the dropping of the higher-order terms is explained in Appendix A.1. The existence of this equation makes a nontrivial prediction on the growth of

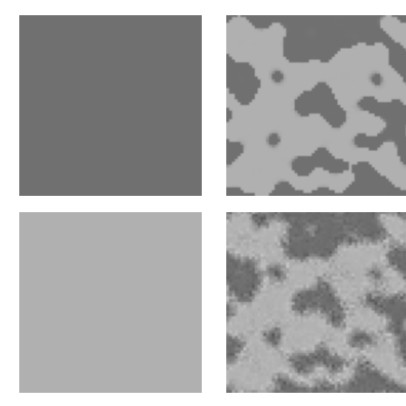

(a) Training samples: grids of $1$ and $-1$.

(b) Results from ELS (top), ConvNet (bottom).

Figure 1: Images generated at end of reverse process by analytical ELS machine and trained ConvNet model.

correlations; in particular, it means that $\rho$ depends entirely on a self-similar, scale-invariant variable

$$\rho(x, y; t) = \rho(|x - y|^2 / \bar{\alpha}_t). \tag{10}$$

A derivation of this result is provided in Appendix A.3. Therefore, at different times, if we plot against the self-similar variable $|x - y|^2 / \bar{\alpha}_t$, we should obtain the same curve for $\rho$.

Overall, during the entire reverse process, we therefore have two distinct regimes. Near the beginning, the diffusive regime dominates, and near the end, no new mutual information is created, and local patches 'snap' to the correct value.

In this case the ELS is easy to formulate, since it has only two possible patches in the training data: all $1$s, or all $-1$s. This leads to an explicit evolution equation of the form:

$$\frac{\partial \phi_t(x)}{\partial t} = \frac{\partial_t \bar{\alpha}_t}{\bar{\alpha}_t} \frac{1}{1 - \bar{\alpha}_t} \left( -\bar{\alpha}_t \phi(x) + \sqrt{\bar{\alpha}_t} \tanh \left( \frac{\sqrt{\bar{\alpha}_t} \sum_{y \in \Omega_x} \phi(y)}{1 - \bar{\alpha}_t} \right) \right) \tag{11}$$

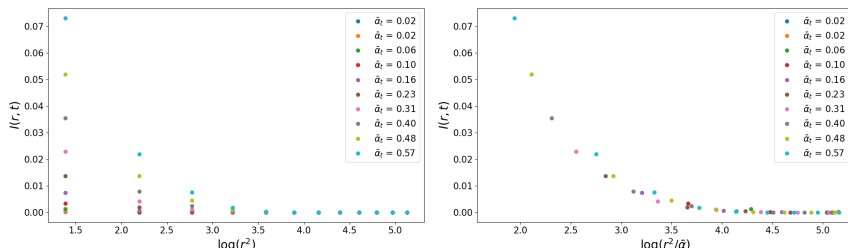

(a) Mutual information curves against position at different times

.8

(b) Mutual information curves against self-similar variable at different times collapse

Figure 2: Verification of diffusive behaviour of ELS machine with $5 \times 5$ kernel size during reverse process.

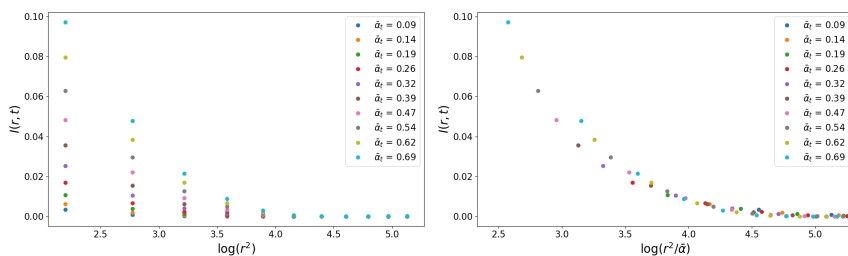

(a) Mutual information curves against position at different times

.8

(b) Mutual information curves against self-similar variable at different times collapse

Figure 3: Verification of diffusive behaviour of learned model during reverse process.

It is possible to also analytically derive from this expression a diffusive regime when $\bar{\alpha}_t$ is small, where the solution's spatial dependence is captured by the self-similar variable $x^2/\bar{\alpha}_t$.

To verify this, we plot the evolution of two-point correlations of the samples in the reverse process under the pure ELS against this self-similar variable in Figure 2. This allows us to see that curves at different timesteps collapse, confirming our theoretical understanding of diffusive scaling.

We also compare the ELS directly to a simple ConvNet trained on an equally split batch of $1$ and $-1$ grids. Extremely similar curve collapse may be observed in Figure 3. Note that we have excluded very small separations where $r$ is a few pixels, as at these scales, the results of the trained model diverge slightly from the pure ELS. This effect may be modelled by changing the size of the locality kernel in the ELS to account for the changing size of the effective receptive field in the ConvNet as in Kamb & Ganguli (2025), but is not required in the present experiment to illustrate our result.

We also carry out the same experiment on a ConvNet trained on MNIST. Similar collapse is observed, confirming our theoretical understanding. For reasons of brevity, the results are shown in Appendix A.4.

# 4 CONCLUSION

We have shown the existence of distinct regimes within the reverse process in this article, together with a quantitative EFT scaling-based description of their dynamics. In particular, in a simple example, we have directly shown the existence of a diffusive regime where information spreads in space according to the self-similar variable $x^2/\bar{\alpha}_t$.

It is interesting to ask whether an effective field theory approach could be helpful for other network architectures, e.g. transformers. It has recently been argued that a notion of locality is inherited from the data independent of the underlying network architecture Lukoianov et al. (2025); we leave this interesting topic for further research.

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

# A APPENDIX

## A.1 TECHNICAL ASPECTS OF DERIVATIVE EXPANSION

Here we focus on an expansion such as (3) near the beginning of the reverse process, where $\bar{\alpha}_t \to 0^+$.

First, note that any term linear in $\phi$ can be absorbed into a field redefinition. For example, if $a_1(\bar{\alpha}_t)\phi$ were such a term, we could redefine $\tilde{\phi} = \exp\left[-\int^t dt' a_1(\bar{\alpha}_t)\right]\phi$ and absorb all resultant factors in the other terms into the definitions of their coefficients.

Note importantly also that such a multiplicative field redefinition has no effect on the mutual information in (5); indeed, in general, if $\tilde{\phi} = \xi(t)\phi$, this has no effect on the expectation value, which is with respect to the initial Gaussian noise, and so

$$\rho_{\tilde{\phi}}(x,y;t) = \frac{\xi(t)^2 \mathbb{E}\left[\phi(x,t)\phi(y,t)\right] - \xi(t)^2 \mathbb{E}\left[\phi_t\right]^2}{\xi(t)^2 \mathbb{E}\left[\phi_t^2\right] - \xi(t)^2 \mathbb{E}\left[\phi_t\right]^2} = \frac{G_t(x,y)}{\mathbb{E}\left[\phi_t^2\right] - \mathbb{E}\left[\phi_t\right]^2} = \rho_\phi(x,y;t) \qquad (12)$$

showing that the mutual information $I(x,y;t)$, which is defined from $\rho(x,y;t)$, remains unchanged.

Next, let us discuss why the higher order terms in such an EFT expansion can be ignored. Central to this argument is the idea of scaling. Near such an endpoint, all physical quantities of interest, such as the mutual information or $\rho(x,y;t)$, become invariant under a defined multiplicative rescaling of the quantities involved.

We can ask how we need to scale $\bar{\alpha}_t, \phi$ if we rescale space $x \to \lambda x$. When $\bar{\alpha}_t \to 0^+$, the endpoint is fixed under multiplicative rescaling $\bar{\alpha}_t \to \lambda^{\Delta_{\bar{\alpha}_t}} \bar{\alpha}_t$, and $\phi$ will also in this limit scale under some power law $\phi \to \lambda^{\Delta_\phi} \phi$.

The question is now which the relevant powers $\Delta_{\bar{\alpha}_t}, \Delta_\phi$ are to achieve this invariance. In the diffusive regime, the scaling is simply $\Delta_{\bar{\alpha}_t} = 2$, irrespective of $\Delta_\phi$. It may be checked that under this scaling, all other terms have higher scaling dimensions, which means they are multiplied by higher powers of $\lambda$, and so are subleading as $\bar{\alpha}_t \to 0$. We therefore arrive at the diffusion equation.

## A.2 LONG-DISTANCE DESCRIPTION OF EQUIVARIANT LOCAL SCORE MACHINE

Here we study the equivariant local score machine discussed in Kamb & Ganguli (2025) and perform a systematic expansion in derivatives and powers of the field $\phi$.

The starting point is Eq (9) of Kamb & Ganguli (2025), which provides the following expression for the score $M_t[\phi](x)$

$$M_t[\phi](x) = \sum_{\varphi \in P_\Omega(\mathcal{D})} \frac{(\sqrt{\bar{\alpha}_t}\varphi(x) - \phi(x))}{1 - \bar{\alpha}_t} W_t(\varphi|\phi, x) \qquad (13)$$

where the weight $W_t(\varphi|\phi, x)$ is defined to be:

$$W_t(\varphi|\phi, x) = \frac{\mathcal{N}(\phi_{\Omega_x}|\sqrt{\bar{\alpha}_t}\varphi, (1 - \bar{\alpha}_t)I)}{\sum_{\varphi' \in P_\Omega(\mathcal{D})} \mathcal{N}(\phi_{\Omega_x}|\sqrt{\bar{\alpha}_t}\varphi'|(1 - \bar{\alpha}_t)I)} \qquad (14)$$

and where the normal distribution can be written explicitly as

$$\mathcal{N}(\phi_{\Omega_x}|\sqrt{\bar{\alpha}_t}\varphi, (1 - \bar{\alpha}_t)I) = n \exp\left[-(1 - \bar{\alpha}_t)^{-1} \sum_{y \in \Omega_x} \left(\phi(y) - \sqrt{\bar{\alpha}_t}\varphi(y)\right)^2\right] \qquad (15)$$

(where the normalization constant $n$ cancels between numerator and denominator, and thus we do not calculate it). Note that the sum over $y$ is over the patch $\Omega_x$ centered about $x$. We have used a notation slightly different from Kamb & Ganguli (2025) in that we write $\varphi(x)$ rather than $\varphi(0)$ to denote the value of the datapoint in question at the center of the patch $\Omega_x$.

We now adapt a continuum notation, i.e.

$$\sum_{y \in \Omega_x} \to \int_{\Omega_x} dy \qquad (16)$$

and systematically expand the right-hand side of Eq. (13) in derivatives of $\phi$ about the evaluation point $x$. This is possible because it depends only on *patches* $\Omega_x$, and is thus a short-range kernel; such a short range kernel always admits an analytic expansion in momentum space and can thus be expanded in derivatives. This expansion is generally only *useful* when the derivatives are themselves small and thus the expansion can be truncated at some order.

Our main point is that this expansion can always be systematically written in terms of an expansion in powers of $\phi$ and derivatives of $\phi$. The first few terms in such an expansion take the form:

$$M_t[\phi](x) = a_{0,0}(\bar{\alpha}_t) + a_{1,0}(\bar{\alpha}_t)\phi(x) + a_{1,1}^\mu(\bar{\alpha}_t)\partial_\mu\phi(x) + a_{1,2}^{\mu\nu}(\bar{\alpha}_t)\partial_\mu\partial_\nu\phi(x) + \cdots \quad (17)$$

where a term in $a_{m,n}$ multiplies $m$ powers of $\phi$ and $n$ derivatives of $\phi$. We will explicitly work out only the first few terms as the expansion rapidly becomes unwieldy.

To proceed we write

$$\phi(y) = \phi(x) + (y-x)^\mu\partial_\mu\phi(x) + \frac{1}{2}(y-x)^\mu(y-x)^\nu\partial_\mu\partial_\nu\phi(x) + \cdots \quad (18)$$

and insert this into the integral in the exponent to find

$$\int_{\Omega_x} dy \left(\phi(y) - \sqrt{\bar{\alpha}_t}\varphi(y)\right)^2$$

$$= \int_{\Omega_x} dy \left(\phi(x) - \sqrt{\bar{\alpha}_t}\varphi(y) + (y-x)^\mu\partial_\mu\phi(x) + \frac{1}{2}(y-x)^\mu(y-x)^\nu\partial_\mu\partial_\nu\phi(x) + \cdots\right)^2 \quad (19)$$

We can now systematically expand in derivatives. We first expand the normal distribution in an expansion as

$$\mathcal{N}(\phi_{\Omega_x}|\sqrt{\bar{\alpha}_t}\varphi, (1-\bar{\alpha}_t)I) = b_{0,0} + b_{1,0}\phi(x) + b_{1,1}^\mu\partial_\mu\phi(x) + b_{1,2}^{\mu\nu}\partial_\mu\partial_\nu\phi(x) + \cdots \quad (20)$$

where the expansion coefficients $b_{m,n}[\bar{\alpha}_t, \varphi]$ implicitly depend on $\bar{\alpha}_t$ and the data point in question $\varphi$. All of the terms without derivatives can be written as:

$$\sum_{m=0}^{\infty} b_{m,0}\phi(x)^m = n\exp\left[-(1-\bar{\alpha}_t)^{-1}\int dy \left[\phi(x) - \sqrt{\bar{\alpha}_t}\varphi(y)\right]^2\right] \quad (21)$$

we may thus read off

$$b_{0,0} = n\exp\left[-(1-\bar{\alpha}_t)^{-1}\int dy \left[\sqrt{\bar{\alpha}_t}\varphi(y)\right]^2\right] \quad (22)$$

etc. and similarly for all $b_{m,0}$. We write down the first few derivative terms that are linear in $\phi$. Explicitly, we have

$$b_{1,1}^\mu = 2b_{0,0}(1-\bar{\alpha}_t)^{-1}\int_{\Omega_x} dy(y-x)^\mu\sqrt{\bar{\alpha}_t}\varphi(y) \quad (23)$$

$$b_{1,2}^{\mu\nu} = b_{0,0}(1-\bar{\alpha}_t)^{-1}\int_{\Omega_x} dy(y-x)^\mu(y-x)^\nu\sqrt{\bar{\alpha}_t}\varphi(y) \quad (24)$$

These expansion coefficients are somewhat analogous to familiar expressions "moments of inertia", where the data variables $\varphi$ play the role of a density.

We now assemble the pieces. For notational convenience denote

$$Z[\bar{\alpha}_t] \equiv \sum_{\varphi'\in P_\Omega(\mathcal{D})} b_{0,0}[\bar{\alpha}_t, \varphi'] \quad (25)$$

The denominator of (14) then becomes

$$\mathcal{N}(\phi_{\Omega_x}|\sqrt{\bar{\alpha}_t}\varphi', (1-\bar{\alpha}_t)I) = Z\left(1 + Z^{-1}b_{1,0}\phi + Z^{-1}\sum_{\varphi'} b_{1,1}^\mu[\varphi']\partial_\mu\phi(x) + Z^{-1}\sum_{\varphi'} b_{1,2}^{\mu\nu}[\varphi']\partial_\mu\partial_\nu\phi(x) + \cdots\right)$$

$$(26)$$

and if we introduce the following notation for a function $a[\varphi]$ of data points $\varphi$:

$$[a]_c \equiv a(\varphi) - Z^{-1} \sum_{\varphi' \in P_\Omega(\mathcal{D})} a(\varphi') \tag{27}$$

then the weight (14) can be written

$$W_t(\varphi|\phi, x) = \frac{1}{Z} \left( b_{00} + [b_{1,0}(\varphi)]_c \phi + [b_{1,1}^\mu(\varphi)]_c \partial_\mu \phi + \cdots \right) \tag{28}$$

Inserting this into (13) and expanding out the terms, we can finally obtain explicit expressions for the first few coefficients in (17):

$$a_{00} = \frac{1}{Z} \frac{1}{1 - \bar{\alpha}_t} \sum_\varphi \sqrt{\bar{\alpha}_t} \varphi b_{00}(\varphi) \tag{29}$$

$$a_{1,0} = \frac{1}{Z} \frac{1}{1 - \bar{\alpha}_t} \sum_\varphi \left( \sqrt{\bar{\alpha}_t} \varphi [b_{1,0}]_c(\varphi) - b_{0,0} \right) \tag{30}$$

$$a_{1,1}^\mu = \frac{1}{Z} \frac{1}{1 - \bar{\alpha}_t} \sum_\varphi \left( \sqrt{\bar{\alpha}_t} \varphi [b_{1,1}^\mu]_c(\varphi) \right) \tag{31}$$

$$a_{1,2}^{\mu\nu} = \frac{1}{Z} \frac{1}{1 - \bar{\alpha}_t} \sum_\varphi \left( \sqrt{\bar{\alpha}_t} \varphi [b_{1,2}^{\mu\nu}]_c(\varphi) \right) \tag{32}$$

These expressions are lengthy and not really very illuminating. The point is really just to state that given a microscopic description (in this case the equivariant local score machine), explicit formulas can be found for the terms in the effective field theory expansion (in this case, writable as sums over the dataset). Furthermore, often symmetries are useful in constraining these terms. For example, imagine that the underlying dataset has a $Z_2$ symmetry which takes $\phi \to -\phi$: if the underlying neural networks are equivariant under this symmetry (Cohen & Welling (2016)), then we can see that $a_{00} = 0$. Similarly, if the dataset has rotational symmetry then we can set $a_{1,1}^\mu = 0$, and $a_{1,2}^\mu = \delta^{\mu\nu}$ as the only rotationally invariant two-tensor.

Finally, the $\bar{\alpha}_t$ scaling of these expressions is of interest: as $\bar{\alpha}_t \to 0$, note that we have $b_{m,n} \sim \sqrt{\bar{\alpha}_t}$ and thus $a_{m,n} \sim \bar{\alpha}_t$, except for $a_{0,0}$ and $a_{1,0}$.

### A.3 DIFFUSION EQUATION EVOLUTION AND SCALING

Here we show that the diffusion equation starting from pure noise leads to self-similar behaviour for the correlation function. Let us use $u(x, t)$ as our field to avoid confusion with the samples $\phi(x, t)$ in the main body of the paper.

We wish to solve

$$\frac{\partial u}{\partial t} = \nabla^2 u$$
$$u(x, 0) \sim \mathcal{N}(0, 1) \tag{33}$$

where the initial condition on the second line indicates that each pixel in the starting pure noise sample is an independent identically distributed standard normal random variable. Note firstly that if we take the expectation of this equation, we see that we start at $\langle u(x, t) \rangle = 0$ and remain there for all time.

Since $u(x, t)$ itself is ultimately random, we want to instead study its statistics, which are deterministic; namely, its two-point correlation function

$$G_t(x, y) := \mathbb{E}\left[ u(x, t) u(y, t) \right] \tag{34}$$

It's important to note that in this context, since the mean $\mathbb{E}[u(x, t)] = 0$ for all $t$, this definition is in this case equivalent to the connected two-point function $\mathbb{E}[u(x, t) u(y, t)] - \mathbb{E}[u(x, t)]\mathbb{E}[u(y, t)]$.

Due to the translational invariance of the grid and the problem, $G_t(x, y)$ must also be translationally equivariant, meaning that it cannot depend on $x, y$ in particular, but only their separation:

$$G_t(x, y) = f(x - y, t) \tag{35}$$

For some function $f$. Let us therefore define the variable

$$z := x - y \tag{36}$$

Now, we can multiply the $x$-evolution equation by $u(y,t)$ and then take the expectation value with respect to the random initial starting grid. This will give us

$$\mathbb{E}\left[\frac{\partial u(x,t)}{\partial t}u(y,t)\right] = \mathbb{E}\left[u(y,t)\nabla_x^2 u(x,t)\right]$$

Now we use the translation equivariance formulated in (35) to realise that on the right hand side, in this case, the derivatives wrt $x$ may be replaced by derivatives in $z$. therefore, the entire equation is

$$\frac{1}{2}\frac{\partial G(z,t)}{\partial t} = \nabla_z^2 G(z,t) \tag{37}$$

We get the factor of $1/2$ on the left hand side since there is another term that comes from the time derivative acting on $u(y,t)$ from the chain rule, which is however identical to the term we already have because of translation equivariance.

Therefore now we want to solve another diffusion equation for the two-point correlation, but now with a deterministic initial condition:

$$G(z,0) = G(x-y,0) = \langle u(x,0)u(y,0)\rangle = \delta(x-y) = \delta(z). \tag{38}$$

Since the pixels were i.i.d. normally distributed.

The easiest way to solve this equation explicitly is a Fourier transform in space, wherein

$$\frac{1}{2}\frac{\partial \tilde{G}(k,t)}{\partial t} = -k^2 \tilde{G}(k,t)$$
$$\tilde{G}(k,0) = 1 \tag{39}$$

Which is straightforward to solve, giving

$$\tilde{G}(k,t) = e^{-2k^2 t} \tag{40}$$

Fourier transforming back to position space finally yields

$$G(z,t) = \sqrt{\frac{\pi}{2t}}e^{-\pi^2 z^2/2t}. \tag{41}$$

In general, the important feature of all such solutions is the distinctive behaviour

$$G(z,t) = g(t)h(z^2/t) \tag{42}$$

where all spatial behaviour is controlled by the *self-similar* variable $z^2/t$. This is a consequence of the fact that in the original diffusion equation, $z$ and $t$ scale in such a way that $z^2/t$ is a dimensionless variable.

## A.4 RESULTS FROM MNIST

For completeness, we reproduce in Figure 4 the samples generated by the periodic CNN architecture diffusion model. We also plot the curve collapse in Figure 5. In this figure, we again plot mutual information at scales beyond $r \sim$ a few pixels. Since the ELS is theoretically most straightforward to formulate with periodic boundary conditions - that is, at the level of the neural network, circular padding - without having to worry about distinguishing edge local patches, we set up our neural network.

However, this results in the neural network learning a characteristic 'digit' scale meaning that the mutual information has a tail, rising to nonzero values at a defined large $r$. This is an artifact of training, and our hypothesis is shown, for $r$ up to this scale, to hold and lead to curve collapse.

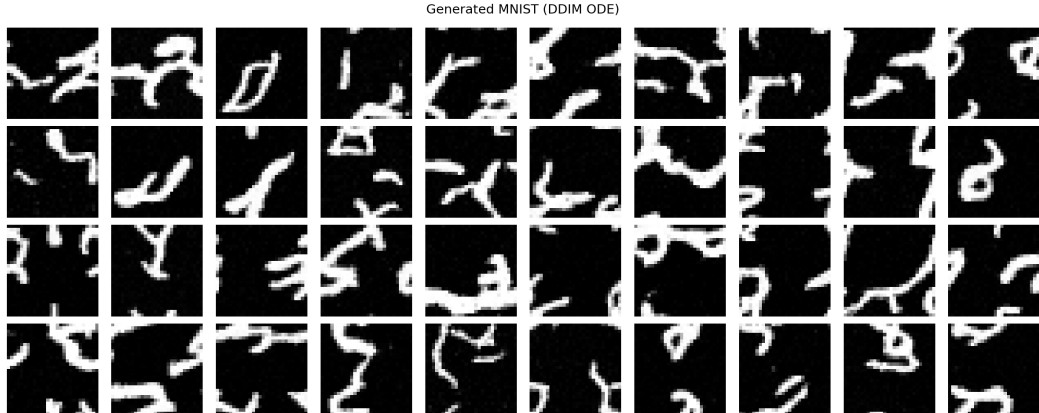

Figure 4: Samples produced by ConvNet with periodic boundary conditions trained on MNIST.

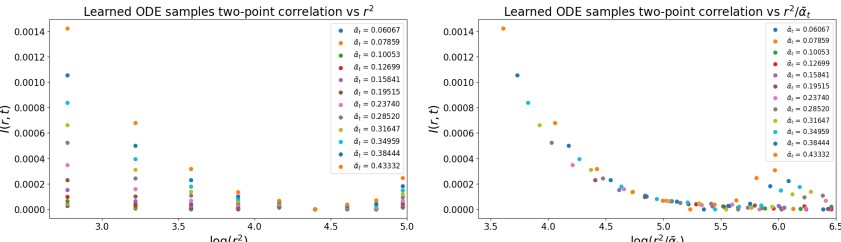

(a) Mutual information curves against position at different times

(b) Mutual information curves against self-similar variable at different times collapse

Figure 5: Verification of diffusive behaviour of ConvNet model trained on MNIST. Due to periodic boundary conditions (formulated to keep ELS theory simple), the network learns a 'digit scale' leading to a tail at large $r$. However, diffusive behaviour is observed up to this $r$ in general.