# OpenReview forum: "Information spreading in diffusion models from effective field theory"
_ICLR.cc/2026/Workshop/Sci4DL — Sci4DL 2026_

### Official Review · Reviewer_jJqe · 2026-02-18

**Fit:** 2
**Significance:** 2
**Confidence:** 3

**Summary:**

This paper provides a study on a simple toy setting to illustrate that Effective Field Theory (EFT) from Physics can be applied to study the dynamics of the reverse process in diffusion models. This work follows the work of [[Kamb and Ganguli, 2025]](https://arxiv.org/abs/2412.20292) closely to demonstrate their point.

**Strengths:**

- The paper is interesting. Albeit, the experiments and results are still at their infancy, but the work does pertain to the workshop.

- Nonetheless, the results do show an interesting and possible reliable trend regarding the Mutual Information on the pixel level for their toy setting.

**Suggestions:**

### Question(s)

- For your result in Fig. (1), why is your generated sample so noisy? Is it because your ConvNet use filters of a very large size?

- I think the results may pertain well to [[Premkumar, 2025]](https://arxiv.org/abs/2409.03817) and [[Biroli et al., 2024]](https://www.nature.com/articles/s41467-024-54281-3) and some of the other works below too --- but the author(s) need to read over these works and see if they can connect their narrative (and results) to them. I think it will give the work a more powerful narrative.

- Your Mutual Information study seems to say that it is quite a stable metric. For example, at $t = 0$, the score function blows up but MI is effectively zero. Anyhow, is there an intermediate phase you can spot with MI or not? Is this stability with the trend of MI primarily due to the fact that your experiments are done on a toy setting?

### Comment(s)
Below, I would like to share some works that could be of interest to you. Much of these works pertain to the study of memorization and generalization in diffusion models, from different perspective. But like [[Biroli et al., 2024]](https://www.nature.com/articles/s41467-024-54281-3), there seems to be an intermediate phase in the transition of these two phases.


- [[Stančević and Ambrogioni, 2026]](https://www.mdpi.com/1099-4300/28/2/195)
- [[Kadkhodaie et al., 2024]](https://arxiv.org/abs/2310.02557)
- [[Bonnaire et al., 2026]](https://arxiv.org/pdf/2505.17638)
- [[Pham et al., 2025]](https://arxiv.org/abs/2505.21777)

---

### Official Review · Reviewer_CWRn · 2026-02-27

**Fit:** 3
**Significance:** 2
**Confidence:** 2

**Summary:**

This study makes partial progress in characterizing the evolution of spatial dependencies along the sampling dynamics of convolutional diffusion models. A local expansion that exploits the locality structure induced by the architecture reveals a diffusive regime where information spreads spatially according to the squared distance divided by (reverse) time--which is verified in experiments on a toy dataset.

**Strengths:**

Developing simplified analysis of sampling in diffusion models can help build intuition about their generalization abilities. Starting from a random initialization without any spatial dependencies, sampling dynamics induce dependencies between pixels. This study offers an analytical approximation to these dynamics (building on Kamb & Ganguli 2025).

**Suggestions:**

Can the first terms of the expansion be given intuitive interpretations?
How does information spread without for the optimal score on the empirical training distribution (no locality)?
How sensitivity is your analysis to the choice of a sampling schedule?
How valid is the Gaussianity assumption and the corresponding mutual information estimation close to the snapping regime?
Is the diffusion regime generic and would the Laplacian term still dominate the dynamics in less toy settings, ie. how robust is the scaling argument?

---

### Meta-Review · Area_Chair_JjSw · 2026-03-01

**Recommendation:** Accept

**Metareview:**

Reviewers stated this work is an interesting extension of  Kamb & Ganguli 2025, and a good fit for the workshop. I recommend accept.

---

### Decision · Program_Chairs · 2026-03-02

Accept